# Wbm0152, an outer membrane lipoprotein of the *Wolbachia* endosymbiont of *Brugia malayi*, inhibits yeast ESCRT complex activity

Lindsay Berardi[1], Alora Colvin[1], Matthew West[2], Greg Odorizzi[2], Vincent J. Starai [1,3]*

**1** Department of Microbiology, University of Georgia, Athens, Georgia, United States of America,
**2** Department of Molecular, Cellular & Developmental Biology, University of Colorado Boulder, Boulder, Colorado, United States of America, **3** Department of Infectious Diseases, University of Georgia, Athens, Georgia, United States of America

* vjstarai@uga.edu

## Abstract

Human pathogenic filarial nematodes of the family Onchocercidae, including *Brugia malayi* and *Onchocerca volvulus,* cause debilitating filarial diseases such as lymphatic filariasis and river blindness. These arthropod-borne pathogens are obligately colonized by the Gram-negative intracellular alphaproteobacterium, *Wolbachia,* which is essential for nematode sexual reproduction, long-term survival, and pathogenicity in the mammalian host. Like many intracellular bacteria, *Wolbachia* likely uses numerous surface-exposed and secreted effector proteins to regulate its ability to persist and replicate within nematode host cells. However, due to the inability to cultivate *Wolbachia* in the laboratory and the genetic intractability of both filarial nematodes and the bacterium, the molecular underpinnings that define the bacterium:nematode relationship are almost completely unknown. In this work, we show that the expression of a *Wolbachia* outer membrane lipoprotein, *w*Bm0152, in *Saccharomyces cerevisiae* inhibits the activity of the Endosomal Sorting Complex Required for Transport (ESCRT), a highly conserved complex essential for autophagy, endosomal maturation, nuclear envelope repair and viral budding in eukaryotic cells. Wbm0152 expression strongly disrupts endosomal maturation, leading to defects in ubiquitylated protein turnover. Using in vivo bimolecular fluorescence complementation, we find that Wbm0152 interacts with the Vps2p subunit of the ESCRT-III subcomplex as well as the Vps2p ortholog (BmVps2, Bm6583b) from a *Wolbachia* host nematode, *Brugia malayi*. These data suggest a novel role of ESCRT in *Wolbachia* persistence, providing insight into the elusive relationship between these two organisms.

**Data availability statement:** Raw data used to generate RapIDeg phenotype scores, Wbm0152:ESCRT colocalization scores, and cryo-EM compartment counts and sizes are available for download via the figshare repository with this link: https://doi.org/10.6084/m9.figshare.30766856.

**Funding:** VJS is supported by a grant from the National Institute of Allergy and Infectious Diseases (R21 AI171573). GO is supported by a grant from the National Institute of General Medical Sciences (R32 GM149202). This research received supplemental funding from the University of Georgia Department of Infectious Diseases. The funders played no role in the study design, data collection and analysis, decision to publish, or in the preparation of the manuscript.

**Competing interests:** The authors have declared that no competing interests exist.

## Author summary

Filarial diseases of mammals, including lymphatic filariasis and canine heartworm, are caused by vector-borne filarial nematodes of the family Onchocercidae. Many of the nematodes in this family are obligately colonized by an intracellular bacterium, *Wolbachia,* which is essential for the nematode's long-term survival, reproduction, and pathogenicity. Therefore, understanding the mechanisms used by *Wolbachia* to persist and replicate within host cells could provide new molecular targets for treating filarial infections. Due to the genetic intractability of both nematode and bacterium, however, significant progress on characterizing these interactions have proven difficult. In this work, we show that a predicted outer membrane lipoprotein, Wbm0152, of the *Wolbachia* endosymbiont of *Brugia malayi*, inhibits yeast Endosomal Sorting Complex Required for Transport (ESCRT) complex activity in vivo. Wbm0152 interacts with a core subunit of the yeast ESCRT-III complex, as well as with the orthologous ESCRT-III protein from *Brugia*. ESCRTs are conserved across eukaryotes and are important for diverse cellular processes such as endosomal maturation, autophagy, and cellular division. As *Wolbachia* persists within a membrane-bound compartment within *Brugia* and must avoid host autophagic pathways, this study presents a potential mechanism by which *Wolbachia* may regulate *Brugia* membrane trafficking pathways to ensure its intracellular survival.

## Introduction

Pathogenic filarial nematodes, such as *Brugia malayi, Wuchereria bancrofti*, and *Dirofilaria immitis*, are a group of parasitic, vector-borne nematodes that are known to cause debilitating and disfiguring illness in millions of humans and animals worldwide. Typically, humans infected with such nematodes are treated with a regimen of anthelmintic drugs such as ivermectin [1–3]. However, due to ivermectin's inability to effectively eradicate adult stage worms in vivo [4], combined with increasing observations of anthelmintic resistance in the nematode population [5–7], the demand for identifying novel drug targets to support the elimination of these nematodes has increased rapidly. Interestingly, filarial nematodes of the family Onchocercidae – which include the genera *Brugia, Wuchereria,* and *Dirofilaria* – are colonized by *Wolbachia,* a Gram-negative, obligately intracellular alphaproteobacterium, which is essential for proper nematode reproduction and survival [8,9]. Elimination of this bacterium from the nematode using doxycycline or tetracycline treatments leads to the sterilization of adult worms, killing of microfilaria, and suppression of infection symptoms in humans [10–12]. Despite the well documented requirement of *Wolbachia* for filarial nematode survival, little is understood about the molecular underpinnings of the bacterial:nematode endosymbiosis. This lack of knowledge predominantly stems from the inability to cultivate *Wolbachia* in the laboratory and the poor genetic tractability of both *Wolbachia* and these filarial nematodes. Therefore, our knowledge about

the *Wolbachia*:nematode relationship stems largely from high-resolution microscopy techniques and heterologous model systems to hypothesize how this bacterium persists within its host.

*Wolbachia*, like many other intracellular bacterial pathogens, including *Legionella pneumophila, Chlamydia trachomatis,* and the closely related alphaproteobacterium *Anaplasma phagocytophilium*, persists and replicates within a host-derived, vacuolar-like compartment [13,14]. The composition of these bacteria-laden compartments is distinct from normal host organelles and they do not fuse with lysosomes, thereby protecting the bacterium from host degradative pathways. Previous research on both nematode-derived and *Drosophila*-derived *Wolbachia* has suggested that this *Wolbachia*-containing compartment is likely formed from ER or Golgi membranes in a manner that does not trigger ER stress [15,16], although the precise lipid and protein composition of this compartment remains unknown. Many intracellular bacteria use dedicated secretion systems to deploy secreted and surface-exposed proteins, termed effectors. These proteins can actively modulate host membrane trafficking and lipid synthesis pathways to create unique intracellular compartments and to inhibit host cell defenses – including phagosome:lysosome fusion – to prevent bacterial degradation [13]. *Wolbachia* contains a Type IV secretion system [17–20] and survives within an intracellular membrane-bound compartment, it is therefore likely that *Wolbachia* also uses secreted effectors to modulate host membrane dynamics, prevents its intracellular degradation, and to ensure the creation of its intracellular replicative niche.

Previously, our laboratory used the unicellular eukaryote *Saccharomyces cerevisiae* as a model system to screen several predicted Type IV-secreted and surface-exposed proteins from the *Wolbachia* endosymbiont of *Brugia malayi* for the ability to manipulate conserved eukaryotic processes [17,21]. One of the candidate effectors tested, *w*Bm0152, was found to belong to the peptidoglycan-associated lipoprotein (PAL) family, a broadly conserved family of proteins found across Gram-negative bacteria, known to be important for stability of the outer membrane, as well as in the regulation of cell division [22–24]. Expression of *w*Bm0152 in yeast led to the aberrant accumulation of an enlarged prevacuolar compartment and the failure to deliver representative endosomal membrane-bound cargo proteins (CPS or Sna3p) to the lumen of the degradative vacuole [17]; these phenotypes are similar to those observed in yeast strains defective in endosomal sorting complexes required for transport (ESCRT) activity [25].

ESCRTs are highly conserved, multisubunit complexes that are responsible for the invagination and scissioning of cellular membranes *away* from the cytosol. In yeast, ESCRTs are essential for intralumenal vesicle formation (ILV) during endosome maturation and for microautophagy [26–29]. In mammalian cells, ESCRTs are important for these same functions as well as membrane repair [30], nuclear envelope remodeling [31,32], and viral budding [33–35]. ESCRT proteins are common targets of intracellular pathogens like *HIV-1, Toxoplasma gondii, and Mycobacterium tuberculosis*, thought to promote the intracellular survival or replication of the pathogen within the host cell [33,36,37]. Despite the well-established connection of host ESCRT activity with intracellular pathogen survival, the requirement of the ESCRT complex has never been directly examined in the context of *Wolbachia's* interaction with its known hosts.

ESCRT subcomplexes assemble in an ordered, stepwise fashion in which the first three complexes (ESCRT-0, -I, -II) function to bind and cluster ubiquitylated proteins on endosomal membranes, as well as recruiting and binding protein subunits of the downstream ESCRT complex. The fourth and most highly conserved complex (ESCRT-III) has the unique job of inducing membrane deformation and scissioning to allow for the creation of the endosomal ILVs. ESCRT-III assembly on endosomes is initiated when myristolated Vps20p is recruited to the endosomal membrane by the assembled ESCRT-II complex (consisting of Vps22p, Vps25p, and Vps36p), which then recruits and initiates the homopolymerization of Snf7p. The Vps2p:Vps24p module of ESCRT-III binds to the Snf7p homopolymer, extending its polymerization laterally, thus creating a three-dimensional helical structure that induces membrane curvature towards the lumen of the endosome [38–43]. The ESCRT accessory protein, Bro1p, also binds directly to Snf7p, promoting the recruitment of the ubiquitin hydrolase, Doa4p, which removes and recycles the ubiquitin moiety from target proteins prior to ILV internalization. Finally, the AAA+ATPase, Vps4p, is recruited to disassemble the ESCRT complexes and completes the scission of the ILV into the lumen of the endosome [44–46]. A model of these protein activities is shown in Fig 1. Disruption of most of the core ESCRT

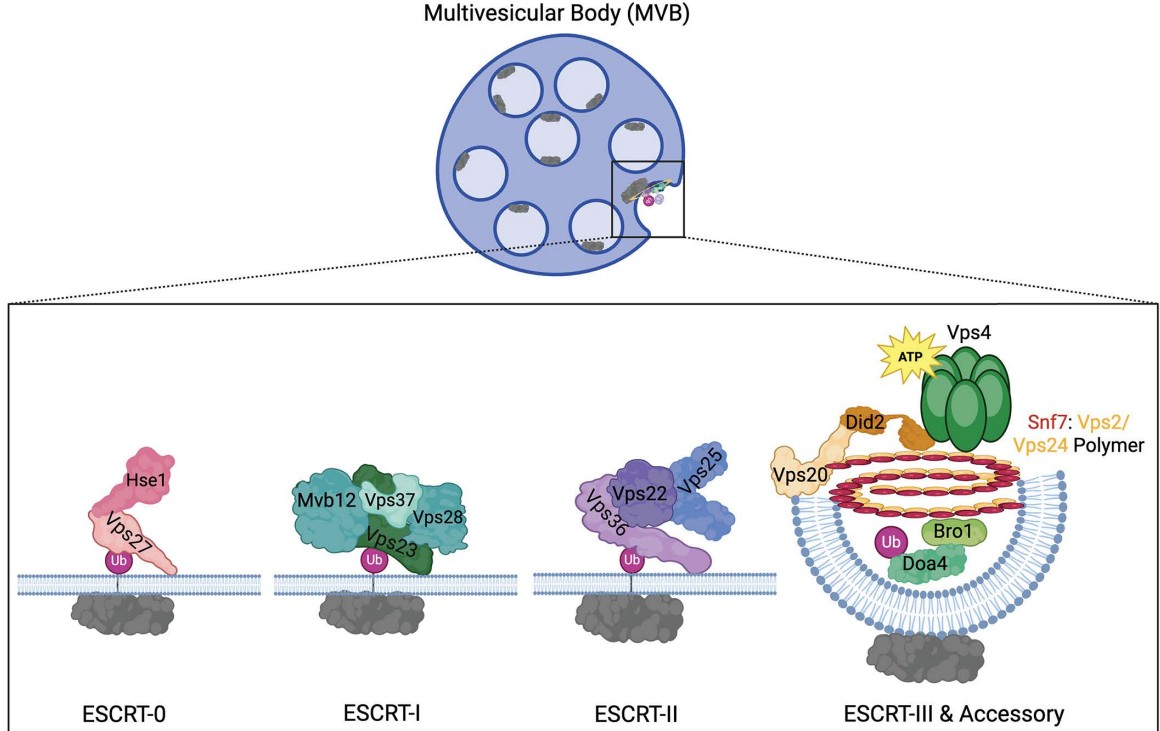

**Fig 1. Simplified model of the ESCRT assembly pathway.** Graphic representation of ESCRT proteins assembling at the endosomal membrane during the formation of endosomal ILVs. ESCRT-0 complex (pink) binds and clusters ubiquitinylated membrane protein (gray) and recruits ESCRT-I (blue/green). ESCRT-I recruits ESCRT-II (purple), resulting in the recruitment of ESCRT-III subunits (orange/yellow). ESCRT-III induces membrane deformation through the creation of the Snf7-Vps2-Vps24 heteropolymer and allows subsequent recruitment of the accessory proteins (green) Bro1p and Doa4p to deubiquitylate target proteins. Lastly, the ATPase Vps4p is recruited to complete ILV formation and disassemble the complex. Diagram created in BioRender. Berardi, L. (2025) https://BioRender.com/9vzuw43.

proteins results in endosomes containing few ILVs and the aberrant accumulation of flattened stacks of endosomes that fail to properly fuse with the vacuole/lysosome; these abnormal endosomes have been termed 'class E compartments' [47].

In this study, we now show that *w*Bm0152 expression inhibits the ILV-formation activity of ESCRT in yeast. By using the rapamycin-induced degradation (RapIDeg) system [48] to visualize ESCRT activity in vivo, we show that *w*Bm0152 expression prevents the ESCRT-dependent translocation of an artificially-ubiquitylated vacuole membrane protein into the lumen of the degradative vacuole. We also demonstrate that expression of *w*Bm0152 inhibits the formation of ILVs in yeast endosomes using cryo-fixation tomography. Through bimolecular fluorescence complementation assays, we also find that Wbm0152 binds to the yeast ESCRT-III protein Vps2p in vivo – as well as the *B. malayi* Vps2p ortholog. Additionally, we show that Wbm0152 likely alters the recruitment of the downstream ESCRT accessory proteins through colocalization studies and electron microscopy. These findings not only identify a bacterial protein capable of inhibiting conserved ESCRT activity in yeast but also suggest an important molecular interaction between *Wolbachia* and its nematode host required to support its intracellular survival.

## Results

### Wbm0152 inhibits ESCRT-dependent protein degradation in yeast

Our previous research showed that expression of *w*Bm0152 in yeast inhibited the normal delivery of the endosomal cargo proteins carboxypeptidase S (CPS) and Sna3p to the degradative vacuole lumen; these phenotypes are also observed in 'class E' protein sorting mutants of yeast caused by defects in Endosomal Sorting Complex Required for Transport (ESCRT)

[17,49,50]. Therefore, we hypothesized that *w*Bm0152 expression may be inhibiting ESCRT either directly or indirectly. To test this hypothesis, we utilized the Rapamycin-Induced Degradation System (RapIDeg) [48] to determine the impact of *w*Bm0152 expression on the ESCRT-dependent vacuolar degradation of an artificially-ubiquitylated protein. In this assay, cells express the vacuolar membrane iron transporter, Fth1p, fused to a GFP and FK506 Binding Protein (FKBP) domain. These strains also harbor the FKBP-rapamycin binding domain (FRB) fused to 3x ubiquitin, allowing the dimerization of the FRB and FKBP domains in the presence of rapamycin, causing the artificial ubiquitylation of the Fth1-GFP protein. This ubiquitylation recruits ESCRT complexes to the vacuole membrane, thus concentrating the Fth1-GFP-3xUb cargo and delivering Fth1-GFP-3xUb into the vacuolar lumen in an ESCRT-dependent manner, leading to the degradation of the Fth1-GFP cargo [48,51].

In empty vector control RapIDeg strains, we observed clear vacuolar membrane localization of Fth1-GFP, as expected (Fig 2A). After rapamycin addition, approximately 90% of the cell population move Fth1-GFP into the vacuole lumen, with concomitant degradation of that protein (Fig 2A and 2B). In contrast, strains expressing *w*Bm0152 failed to degrade Fth1-GFP with only 2% of cells accumulating Fth1-GFP in the vacuole lumen 2h after treatment with rapamycin (Fig 2A–2C). Expression of the *w*Bm0152 ortholog from *E. coli* (*pal*, S1A Fig) in these strains did not impact the ability of yeast ESCRT to package and mobilize Fth1-GFP into the vacuole lumen for degradation (Fig 2A–2C), despite being expressed to similar levels as Wbm0152 in vivo (S1B Fig). Therefore, *w*Bm0152 expression in yeast induces an ESCRT inhibition phenotype that is not caused by general overexpression of a foreign bacterial PAL family protein.

## Expression of *w*Bm0152 in yeast induces ESCRT mutant-like growth defects

Previously, our lab observed a slight growth defect of *w*Bm0152-expressing yeast strains in the presence of zinc and caffeine, suggesting expression of Wbm0152 may alter endolysosomal membrane compartment function [17]. Based on our RapIDeg results indicating that *w*Bm0152 likely inhibits ESCRT activity, we sought to determine if *w*Bm0152 expression induced specific growth phenotypes like those observed in yeast strains lacking individual ESCRT subunit proteins. Previous research identified that the 1,3-β-glucan-binding azo dye, Congo red [52], inhibited the growth of several individual ESCRT subunit mutants [53]. This same study also revealed that many of the ESCRT-III or ESCRT-III related subunits have distinct levels of growth inhibition when grown on media containing Congo red. Particularly, ESCRT-III mutant strains such as *vps2Δ* and *vps24Δ* are exquisitely sensitive to the presence of Congo red, while other strains – like *snf7Δ* and *vps20Δ* – are only moderately sensitive to these conditions, allowing us to compare growth patterns between ESCRT subunit mutants and *w*Bm0152 expressing strains.

Vector control strains are insensitive to the presence of 15 μg ml$^{-1}$ Congo red on minimal medium (Fig 3). Strains constitutively expressing *w*Bm0152, however, are strongly inhibited for growth under these conditions; expression of *E. coli pal* does not increase the sensitivity of yeast to Congo red (Figs 3 and S1C). When comparing the growth of *w*Bm0152-expressing strain to representative deletion mutants of ESCRT complex subunits, we noted that this strain has similar levels of sensitivity to Congo red as *vps2Δ*, *vps24Δ,* and *bro1Δ* strains; no other ESCRT subunit deletion strain showed comparable growth sensitivities under these conditions. These results indicate that expression of *w*Bm0152 likely impacts the function of the yeast ESCRT-III subcomplex, as well as the downstream ESCRT accessory proteins (Bro1p) required for efficient ESCRT function. Vps2p and Vps24p are known to interact in a bipartite module, binding to polymerized Snf7 to promote the creation of the 3-dimensional helix to induce endosomal membrane deformation and ILV formation [38]. Bro1p, known to be activated by the polymerization of Snf7, is a key accessory protein required for the recruitment of the ubiquitin hydrolase, Doa4p [54]. In addition to its role in recruiting Doa4p to ESCRT-III, Bro1p:Snf7p interactions also inhibit the association of Vps4p with the ESCRT complex, allowing time for the deubiquitylation of protein cargo by Doa4p [55]. Therefore, Wbm0152 may be disrupting the recruitment of the downstream ESCRT accessory proteins to the assembled complex. As ESCRT-III is primarily responsible for the physical deformation of the endosomal membrane to generate ILVs, we next chose to directly observe the impact of *w*Bm0152 expression on yeast endosomal maturation via electron microscopy.

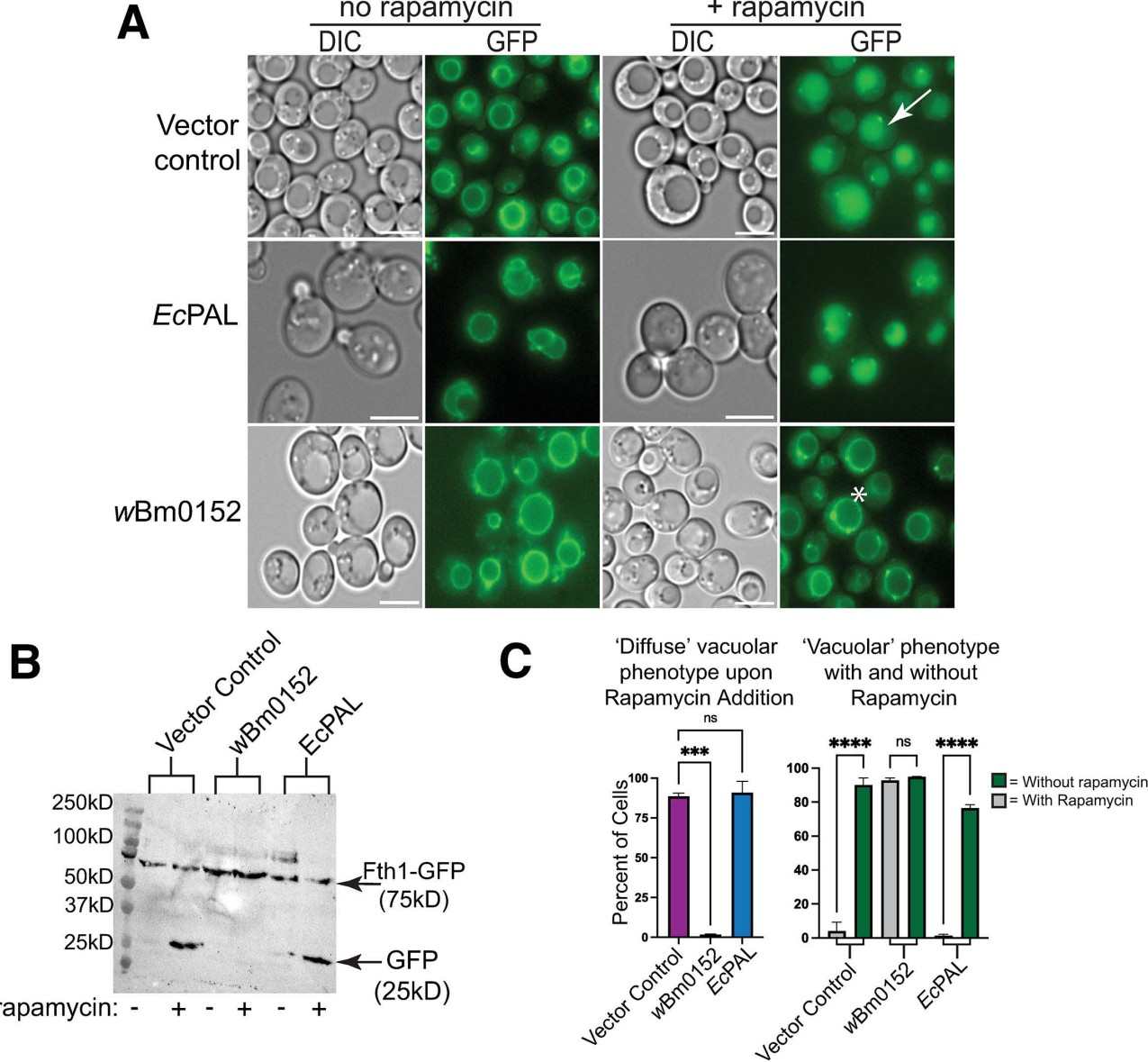

**Fig 2. Wbm0152 expression inhibits ESCRT-dependent ubiquitylated protein turnover.** RapIDeg yeast strains harboring either a vector control or the β-estradiol inducible *w*Bm0152 or *Ec*PAL expression vectors were assayed for vacuolar turnover of Fth1-GFP-FKBPx2, as in Methods. **A)** Representative images showing intracellular localization of Fth1-GFP in response to 1 μg mL$^{-1}$ rapamycin; bar = 5 μ. Vacuolar localization of Fth1-GFP counted as either 'diffuse' lumenal localization (white arrow) or 'vacuolar' membrane localization (asterisk). **B)** Representative α-GFP immunoblot of whole cell lysates prepared from cells in **A)**.**C)** Results of One-Way ANOVA performed on percentage of cells displaying indicated phenotype in each condition compared to the empty vector control. (****) $p \leq 0.0001$; (***) $p \leq 0.001$; (ns) not significant; n > 300 cells for each condition.

## wBm0152 expression results in defective ILV formation and endosomal morphology

During endosomal maturation, ESCRT activity is required on early endosomal compartments to create the ILVs indicative of late endosomes (also termed multivesicular bodies, or MVBs). The formation of endosomal ILVs are necessary to effectively degrade ubiquitylated transmembrane cargo proteins on endosomes, as these transmembrane proteins would otherwise remain embedded in the vacuolar limiting membrane after endosome:vacuole fusion and fail to be

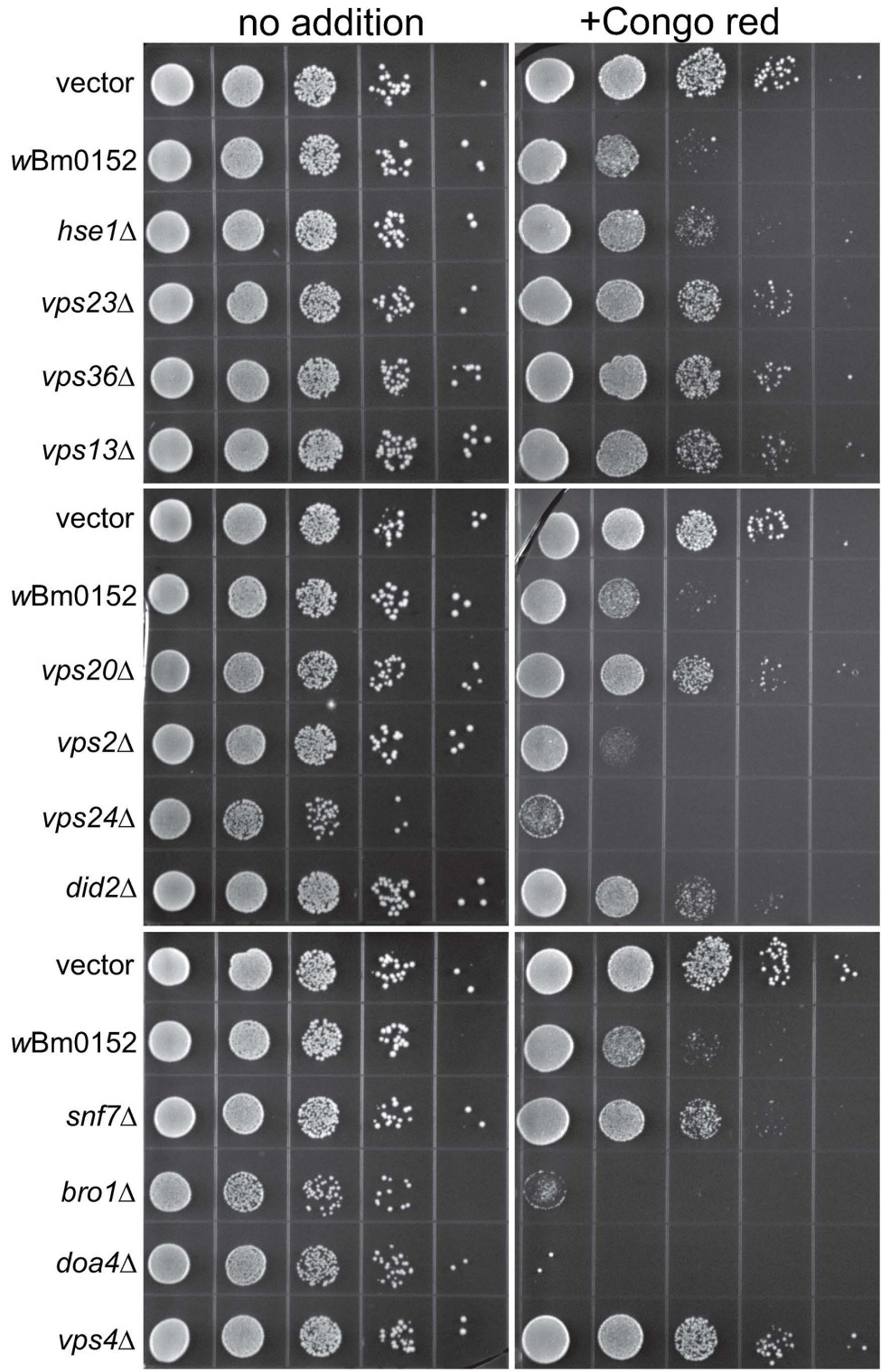

**Fig 3. Wbm0152 expression phenocopies ESCRT mutant growth defects.** BY4742 yeast strains harboring the indicated ESCRT subunit deletion, an empty vector control, or the constitutively expressing pYES$_{TDH3}$-wBm0152 (Methods) were grown overnight at 30° C in CSM-URA media. Cultures were diluted to a final OD$_{600}$ = 1.0, serially diluted 10-fold four times into sterile water, and 10 μL of each dilution was spotted to CSM media lacking uracil either lacking or containing 15 μg mL$^{-1}$ Congo red. Plates were incubated at 30° C and imaged after 72h.

delivered to vacuolar proteases. To visualize to impact of wBm0152 expression on endosomal maturation, we used cryo-fixation for electron tomography to generate high-resolution images of endolysosomal membrane compartments and associated organelles; cells fixed under these conditions retained clear membrane-bound compartments (S2 Fig). We observed spherical MVBs in both the control strain and the wBm0152-expressing strain, although the strain harboring wBm0152 appeared to have a slight reduction in the numbers of MVBs per 100 cell profiles (Fig 4A–4F, quantified in 4G). Notably, we also found tubular MVBs in strains expressing wBm0152 (Fig 4E and 4F), similar to those found in other ESCRT-impaired yeast strains like *vta1Δ, did2Δ, vps60Δ,* or *BRO1*-overexpressing strains [45,46,55–61]. Movies showing membrane structures through the tomographic spaces used for modeling these MVBs can be seen in Supplemental Information (vector control, S1 Movie; wBm0152, S2 Movie). Furthermore, we noted numerous aberrant membrane structures in wBm0152-expressing cells when compared to control strains, including tubular endoplasmic reticulum and increased frequency of lipid droplets (Fig 4E and 4F). Altered endoplasmic reticulum morphologies and increases in intracellular lipid droplet concentrations are phenotypes that have been previously observed in ESCRT mutants, as both lipid droplet consumption [62] and microautophagy of the ER [63] require ESCRT activity for delivery into the vacuole.

To better observe the morphology of the smaller and less frequent MVBs found in wBm0152-expressing strains, we modeled the membrane compartments found in each tomographic stack and calculated the physical dimensions of the relevant structures. Compared to control strains, the endosomes in strains harboring wBm0152 contained fewer ILVs per MVB (Fig 4G), although those infrequent ILVs tended to be larger in diameter than those found in control strain MVBs (33.38 nm *vs* 26.51 nm, $P<0.0001$; Fig 4H). By calculating the surface area of the limiting membranes of the MVBs and ILVs in both strains, we found that the cumulative surface area of the MVBs in wBm0152-expressing strains was approximately twice that of the control strains with a concomitant reduction in ILV membrane surface area (Fig 4I). Therefore, wBm0152 expression strongly inhibits the formation of endosomal ILVs in vivo, providing additional evidence of its ability to inhibit ESCRT complex activity.

## Wbm0152 colocalizes with representative ESCRT complex subunits

To determine if Wbm0152 localizes with ESCRT protein subunits in vivo, we performed a colocalization analysis with GFP-tagged candidate ESCRT protein subunits and Wbm0152-mRuby. Each ESCRT-GFP subunit (Vps27-GFP, ESCRT-0; Vps36-GFP, ESCRT-II; Snf7-GFP, ESCRT-III; Bro1-GFP, accessory) was localized to multiple punctate structures indicative of endosomal compartments, as expected [39,54,64,65] (Fig 5A). In strains co-expressing the indicated ESCRT-GFP protein with the wBm0152-mRuby expression vector, strong colocalization of Wbm0152-mRuby with Vps27-GFP, Vps36-GFP, and Snf7-GFP was observed. (Fig 5A and 5B). Interestingly, however, Wbm0152 failed to strongly colocalize with the ESCRT accessory protein Bro1-GFP (Fig 5A). To quantify these colocalization interactions between Wbm0152 and ESCRT subunits, we also visualized the colocalization of a known interacting protein pair, Snf7-mRuby and Bro1-GFP [55], and a non-interacting protein pair, Snf7-mRuby and the Golgi mannosyltransferase subunit, Mnn9-GFP [66] (Fig 5A). The Pearson correlation coefficient was determined for each of these interactions, which revealed that the interactions between Wbm0152 and the ESCRT proteins Vps27p, Vps36p, and Snf7p were statistically different from the negative control (Snf7-mRuby:Mnn9-GFP), while the colocalization score between Wbm0152 and Bro1p was not significantly different than the negative control (Fig 5B), thus showing that Wbm0152 localizes with core ESCRT subunits, but not the downstream ESCRT accessory proteins. This finding is especially interesting, as it is known that Bro1p normally binds to – and colocalizes with – Snf7p, as we observe in the positive control, to direct the recruitment of Doa4p to the assembled ESCRT complex [54,55]. Therefore, it is possible that Wbm0152 disrupts the recruitment of Bro1p to the assembled ESCRT complex.

## Wbm0152 interacts with ESCRT-III protein Vps2p

As it appears that wBm0152 expression either inhibits the activity of ESCRT-III or disrupts ESCRT complex assembly dynamics, we hypothesized that Wbm0152 would interact with at least one protein subunit of the ESCRT-III complex. By

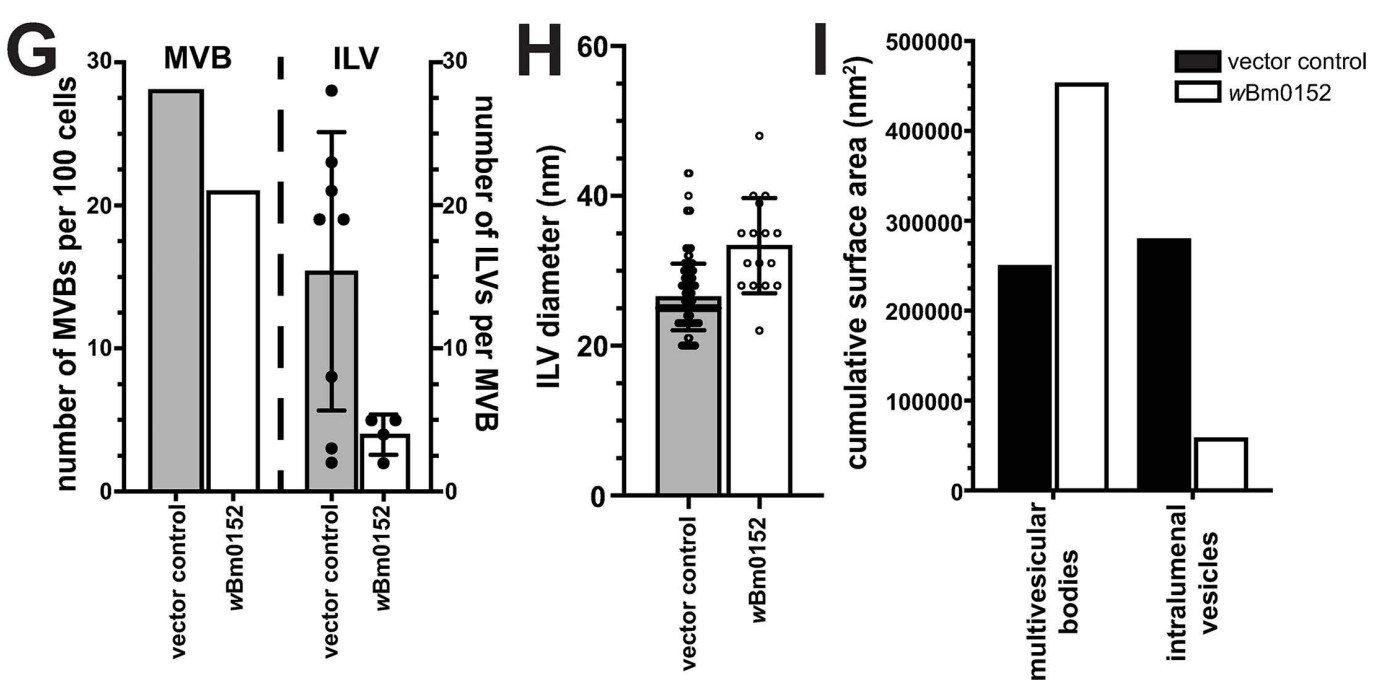

**Fig 4. *w*Bm0152 expression inhibits intralumenal vesicle formation in vivo.** Yeast strains harboring either the vector control (A-C) or *w*Bm0152 expression vector (D-F) were visualized via cryo-fixation electron tomography (Methods, bar = 100 nm) and membrane structures were modeled as indicated: MVBs, yellow; degradative vacuole, red; endosomal ILVs, red spheres; ER, green; lipid droplets; white. Raw images labeled to show degradative Vacuole (V), MVBs (MVB) and lipid droplets (LD). G) Quantification of total MVBs (left) or ILVs per MVB (right) observed across 100 cell profiles for the indicated yeast strain. H) Calculated diameters of all ILVs observed in cell profiles from (G). I) The total surface area of the limiting membranes of observed MVBs or ILVs from the tomogram profiles of the indicated strains were quantified, as outlined in Methods.

utilizing a split-Venus bimolecular fluorescence complementation assay (BiFC) [67–69], we sought to identify potential ESCRT binding partners of Wbm0152 through the reconstitution of fluorescence upon protein:protein interactions. Therefore, yeast strains harboring individual ESCRT protein fusions with either the non-fluorescent Venus N-terminal (VN) or Venus C-terminal (VC) domains were transformed with a copper-inducible expression construct containing *w*Bm0152 fused to the complementary Venus fragment. Should Wbm0152 interact with any of these ESCRT subunits and bring the VN and VC domains into close proximity, the fluorescent Venus chromophore will be reconstituted and detectable via fluorescence.

In yeast strains expressing Snf7-VC and Vps2-VN – proteins from the ESCRT-III complex known to directly interact in vivo [45] – we detected numerous fluorescent punctae in both the induced and uninduced conditions. The presence of punctae in the uninduced conditions can be explained by 'leakiness' in the *CUP1* promoter, likely causing low levels of expression of our construct even in the absence of inducer (Fig 6). Importantly, these punctae were not observed in strains harboring the Snf7-VC and empty vector control constructs (Fig 6). Using the inducible *w*Bm0152-VN (or-VC) expression construct from above and a number of ESCRT subunit strains expressing the reciprocal split-Venus fusion, we only identified fluorescent punctae in the Vps2-VN strain harboring the *w*Bm0152-VC construct (Fig 6); no other strains tested showed fluorescence under similar conditions (S3 Fig; tested Vps25p (ESCRT-II), Vps20p (ESCRT-III), Vps24p (ESCRT-III), Snf7p (ESCRT-III), Bro1p (accessory), the Snf7p-binding domain of Bro1p (BOD [42]), and Doa4p). Interestingly, we did not see reconstitution of Venus fluorescence in Vps2-VC/*w*Bm0152-VN strains (S3 Fig). This observation could be due to the C-terminal -VN fusion forcing Vps2 into a constitutively active, 'open' conformation to bind Wbm0152, which is known to occur with yeast and mammalian ESCRT-III GFP fusion proteins [37,43,70]]. This forced 'open' confirmation by the VN tag encourages interactions amongst other ESCRT subunits—as the C terminus must be in this confirmation order to expose the first two alpha helices of the ESCRT-III proteins to allow for interaction [43]. Therefore, this forced 'open' confirmation is likely occurring in each of the tagged ESCRT-III subunits in this assay and observing only Wbm0152:Vps2 binding with the 'open' confirmation of Vps2 – and not other ESCRT-III subunits – suggests that this interaction between Wbm0152 and Vps2 is specific amongst the ESCRT-III subunits. Alternatively, the VC/VN components in this putative Vps2p:Wbm0152 interaction may simply be in the wrong orientation to form the fluorescent chromophore upon interaction. Unfortunately, ESCRT-III subunits interact via their N-termini [38], so moving the VN tag to this side of the protein would likely lead to significant disruption of ESCRT complex formation. Regardless, these results show that *Wolbachia* Wbm0152 likely interacts specifically with the Vps2p ESCRT-III subunit in vivo.

## Wbm0152 requires normal ESCRT assembly for activity

Taking advantage of the growth defect we observed on media containing Congo red upon *w*Bm0152 expression in a wildtype strain background, we sought to determine which ESCRT components were required for the Wbm0152-mediated toxicity on this medium. Therefore, candidate ESCRT deletion strains representing one protein from each of the ESCRT-0,-I, and -II complexes – as well as each subunit of the ESCRT-III and downstream accessory complexes – were grown on media containing Congo red with and without *w*Bm0152 expression. As we observed previously, strains missing components of the ESCRT-0, -I, and -II complexes harboring a vector control did not show strong growth defects in the presence of Congo red, when compared to the vector control wild type strain (Fig 7). Strikingly, *w*Bm0152 expression in

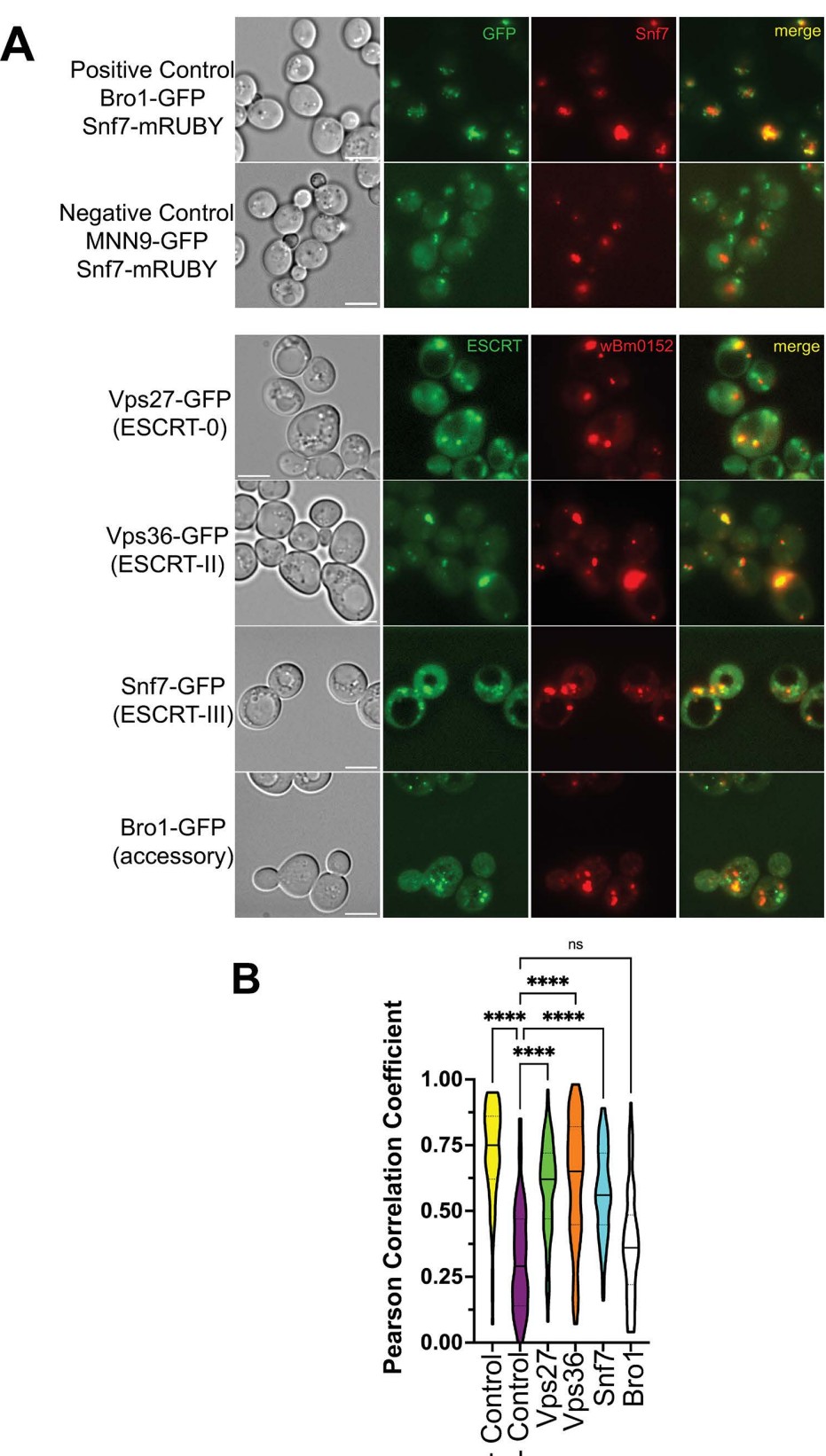

**Fig 5. Wbm0152 colocalizes with core ESCRT subunits in vivo.** A) SEY6210 yeast strains harboring the indicated chromosomal ESCRT subunit-GFP or Mnn9-GFP and the pYES-wBm0152-mRuby or pYES-Snf7-mRUBY expression vector were grown to saturation at 30° C in selective media. Cells were diluted 1:10 into fresh selective media containing 1 μM β-estradiol, outgrown for 6h at 30° C, and imaged. Bar = 5 μ. B) Truncated violin plot showing calculated Pearson Correlation Coefficient of the positive (Bro1/Snf7 colocalization) and negative controls (Mnn9/Snf7) colocalization along with the scores for the indicated ESCRT:wBm0152 colocalization. One-Way ANOVA analysis with the negative control PCC scores as the reference group was run. (****) $p \leq 0.0001$; (ns) not significant; $n \geq 100$ cells for each condition.

these mutant backgrounds did not cause additional growth defects on Congo red, in contrast to the wild type (Fig 7). Similarly, wBm0152 expression did not induce Congo red growth defects in vps20Δ or snf7Δ strains (Fig 7). As each ESCRT subcomplex generally does not assemble and accumulate on the endosomal membrane without the proper assembly and recruitment of the previous ESCRT subcomplex [43,64,65,71,72], these results show that normal, ordered ESCRT complex assembly is required for wBm0152 activity under these growth conditions. ESCRT disassembly was not important for the toxicity of wBm0152 expression, however, as vps4Δ strains remained sensitive to wBm0152 expression on Congo red (Fig 7). Due to the strong growth defect observed on Congo red in vps2Δ, vps24Δ, bro1Δ, and doa4Δ strains alone, the necessity of these subunits for Wbm0152 toxicity could not be determined. Taken together, these data suggest the action of Wbm0152 requires fully assembled ESCRT complex and acts upstream of the vps4Δ disassembly apparatus.

### Wbm0152 interacts with the *Brugia* Vps2 ortholog in yeast

We initiated this work to identify an interaction between a *Wolbachia* candidate effector protein, Wbm0152, with a conserved eukaryotic protein from yeast, which would help elucidate the activity of Wbm0152 in the *Wolbachia*:*Brugia malayi* endosymbiosis. As ESCRT complexes are broadly conserved across eukaryotes, we identified the *VPS2* ortholog (Fig 8A) from the *B. malayi* genome (*BM6583*, isoform b) [73] and constructed a yeast codon-optimized expression vector to assess Wbm0152:Bm6583b interactions in vivo.

As it is known that Vps2p interacts directly with Snf7p during ESCRT-III complex formation and that interaction can be observed using BiFC, we first ensured that Bm6583b (hereafter, *Bm*Vps2) would interact with yeast Snf7p in vivo. As observed previously, a yeast strain expressing Snf7-VC and Vps2-VN reconstitutes Venus fluorescence (Fig 8B), confirming the expected Vps2p:Snf7p interaction in vivo. Expression of *Bm*Vps2-VN from the endogenous yeast *VPS2* promoter in the Snf7-VC background also produced fluorescent punctae (Fig 8B), showing that *Bm*Vps2 also interacts with yeast Snf7p and likely engages in ESCRT-III complexes with the yeast protein subunits. Next, we introduced the inducible wBm0152-VC vector previously used into strains expressing *Bm*Vps2-VN in both wild type and vps2Δ backgrounds. In both backgrounds, we observed the reconstitution of Venus fluorescence upon induction of wBm0152-VC expression (Fig 8C), showing that wBm0152 interacts with *Bm*Vps2 in a similar manner to yeast Vps2p in vivo, thus strengthening the possibility that Wbm0152 interacts with ESCRT-III subunits in the nematode.

### Discussion

In this study, we have shown that expression of Wbm0152 in yeast inhibits the eukaryotic ESCRT complex, potentially via interactions with the core ESCRT-III protein, Vps2p. This inhibitory activity requires the formation of a properly assembled ESCRT complex, as in the absence of ESCRT-0, -I, -II, and most -III subunits, wBm0152 expression failed to induce growth defects on media containing Congo red. Importantly, Wbm0152 was also found to interact with the *Brugia* Vps2 ortholog in yeast (Bm6583b), providing evidence of a physiologically-relevant interaction within the nematode that cannot be easily confirmed in the natural host. These data highlight a potential role for Wbm0152 in the regulation of *Brugia* ESCRT activity during endosymbiosis.

Despite the many advances made in the analysis of *Wolbachia*:host interactions, much of the current research focuses on understanding *Wolbachia* survival and transmission in arthropod hosts, in which the relationship between the

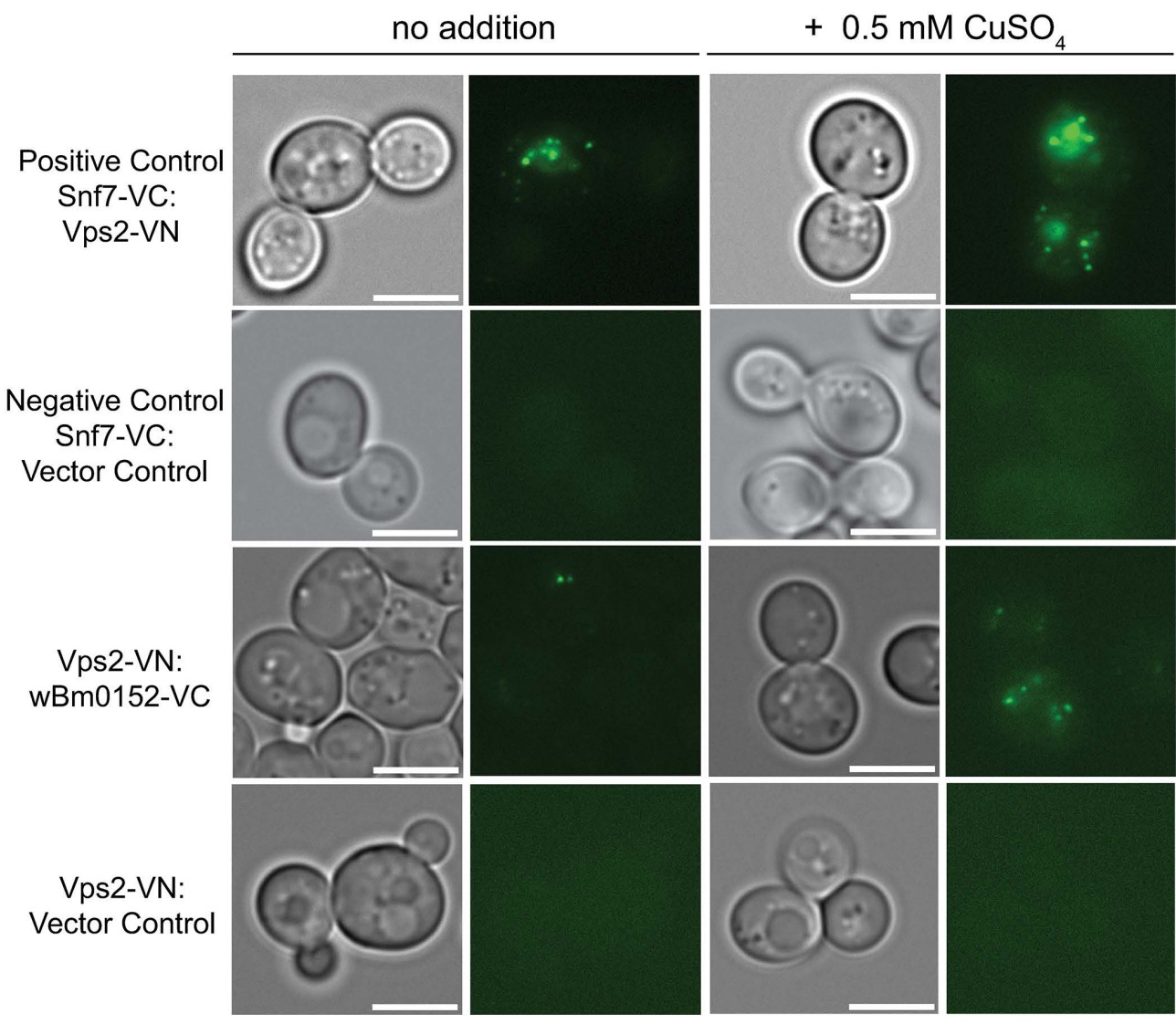

**Fig 6. wBm0152 binds ESCRT-III subunit Vps2 in vivo.** SEY6210 strains harboring either the N-terminus (VN) or C-terminus (VC) of a Venus-YFP molecule on the C terminus of the indicated ESCRT subunit were transformed with the corresponding copper-inducible pYESCUP1-wBm0152-VN or pYESCUP1-wBm0152-VC plasmid. Strains were grown in CSM media lacking uracil for 18h at 30° C with shaking, diluted 1:10 into fresh selective media lacking or supplemented with 0.5 mM CuSO4, and outgrown for 6 hours before imaging. Bar = 5 μ; images are representative of three separate experiments.

bacterium and host is usually parasitic in nature. Through a combination of mutant insect lines and cell culture systems, researchers have shown the importance of host actin dynamics for the uptake and maternal transmission of *Wolbachia* during host development [74,75] as well as host microtubule dynamics to drive the asymmetic segregation of *Wolbachia* in developing embryos, [76,77] although information regarding *Wolbachia's* ability to actively manipulate these host pathways remains lacking. Understanding of the *Wolbachia*:nematode relationship, however, remains even less clear due to the genetic intractability of *Brugia* and *Wolbachia*, the general difficulty to rear such nematodes, the lack of primary or immortalized nematode cell lines, and the fact that nematode-derived *Wolbachia* (supergroups C and D) is known to express evolutionarily-divergent Type IV-secreted effectors than the arthropod-derived *Wolbachia* (supergroups A and

**Fig 7. Wbm0152 activity requires ESCRT assembly.** BY4742 yeast strains harboring the indicated ESCRT subunit deletion, an empty vector control, or the constitutively expressing pYES$_{TDH3}$-wBm0152 (Methods) were grown overnight at 30° C in CSM-URA media. Cultures were diluted to a final OD$_{600}$ = 1.0, serially diluted 10-fold four times into sterile water, and 10 µL of each dilution was spotted to CSM media lacking uracil either lacking or containing 15 µg mL$^{-1}$ Congo red. Plates were incubated at 30° C and imaged after 72h.

B) [78]. Therefore, the development of an alternative biological model system for the study of these *Wolbachia* effector proteins found in filarial nematodes is demanded. Our lab, as well as several others, have had great success utilizing *Saccharomyces cerevisiae* as a model eukaryotic cell to study the molecular underpinnings of many host:bacterium interactions, especially when the target of these bacterial effectors is conserved in eukaryotes [79–81].

Many intracellular bacteria such as *Salmonella enterica* and *Mycobacterium tuberculosis* are known to directly modulate host ESCRT activity, likely to support intracellular survival and transmission. Although there are well documented ESCRT- interacting proteins, like the *Mycobacterium* secreted effectors EsxG and EsxH [82], ESCRT-interacting domains amongst intracellular bacteria are not well conserved, nor are their molecular activities on ESCRT well known. Despite the lack of understanding on how intracellular bacteria interact with ESCRT, there is a wealth of knowledge on how many viruses, including HIV-1, interact with and manipulate ESCRT to promote viral budding within mammalian cells. Most of these viral effector proteins are known to contain P(S/T)AP, PPxY, or YPx(n)L motifs that mimic motifs present on ESCRT [83]. The P(S/T)AP domain is known to interact with TSG101, the human ortholog to the yeast ESCRT-I protein Vps23p [84–86]. The PPxY motif is noted to interact with NEDD4, the human ortholog to the yeast Rsp5p protein [87–89], which is a ubiquitin ligase that interacts with Vps27p and Hse1p to regulate the biogenesis of MVBs [90,91]. Lastly, the YPX(n)L motif is known to interact with ALIX (yeast Bro1p), [64] which will also be found in complexes containing the ESCRT-III subunits CHMP2, (Vps2p ortholog), CHMP4 (Snf7p ortholog), and Vps4p (reviewed in [92]). Some of these domains have been found on ESCRT-interacting proteins from other intracellular pathogens, such as GRA14 and RON4 from *Toxoplasma gondii*, which are known to contain the P(T/S)AP and YPX(n)L domains, respectively [36,93]. However, it is important to note that *Toxoplasma* has other known ESCRT-interacting proteins, such as GRA64, which is noted to interact with ESCRT subunits TSG101, VPS37, VPS28, UMAD1, ALG-2, and CHMP4, but does not contain any of the known ESCRT-binding motifs previously mentioned [94]. Therefore, it is difficult to predict the ESCRT-binding activity of a protein based on sequence information alone.

Previous studies have noted the knockdown of ESCRT components like the Vps2 homolog, CHMP2, and the SNF7 homolog, SHRB, in JW18 *Drosophila melanogaster* embryonic cell lines caused an increase in *Wolbachia* populations in vivo [95], suggesting that manipulation of host ESCRT complex activity by the bacterium is essential for its persistence within the host cell. Additionally, activation of autophagic pathways via exogenous addition of rapamycin to either *Wolbachia*-infected *Drosophila* cell lines or *Brugia malayi* was shown to reduce intracellular *Wolbachia* populations, presumably through lysosomal degradation of the bacterium [96]. As ESCRT activity is required for both macro and microautophagy [63,97,98], it is possible that Wbm0152 (and other *Wolbachia* orthologs) dampens host ESCRT activity to prevent autophagic degradation of the bacterium. Furthermore, RNA-Seq data obtained from 6-week *Brugia malayi* microfilaria isolated from tetracycline-treated animals to eliminate *Wolbachia* from the nematode, found that the transcription of *Brugia* genes producing proteins involved in multivesicular body formation were upregulated in the absence of *Wolbachia* [99], indicating that the presence of *Wolbachia* does appear to alter *Brugia*/filarial nematode endolysosomal membrane dynamics. *w*Bm0152 is highly expressed in adult female nematodes along with other T4SS predicted effectors like *w*Bm0490, a gene encoding a protein with homology to BAX inhibitors, known to be important for the suppression of apoptosis [100]. Wbm0152 peak expression coinciding with other proteins suggested to manipulate eukaryotic cellular biology further implies its role in the manipulation of host cell function. Furthermore, its increased expression in adult female worms – as opposed to adult males – suggests a potential role of *w*Bm0152 in the establishment of *Wolbachia* within developing

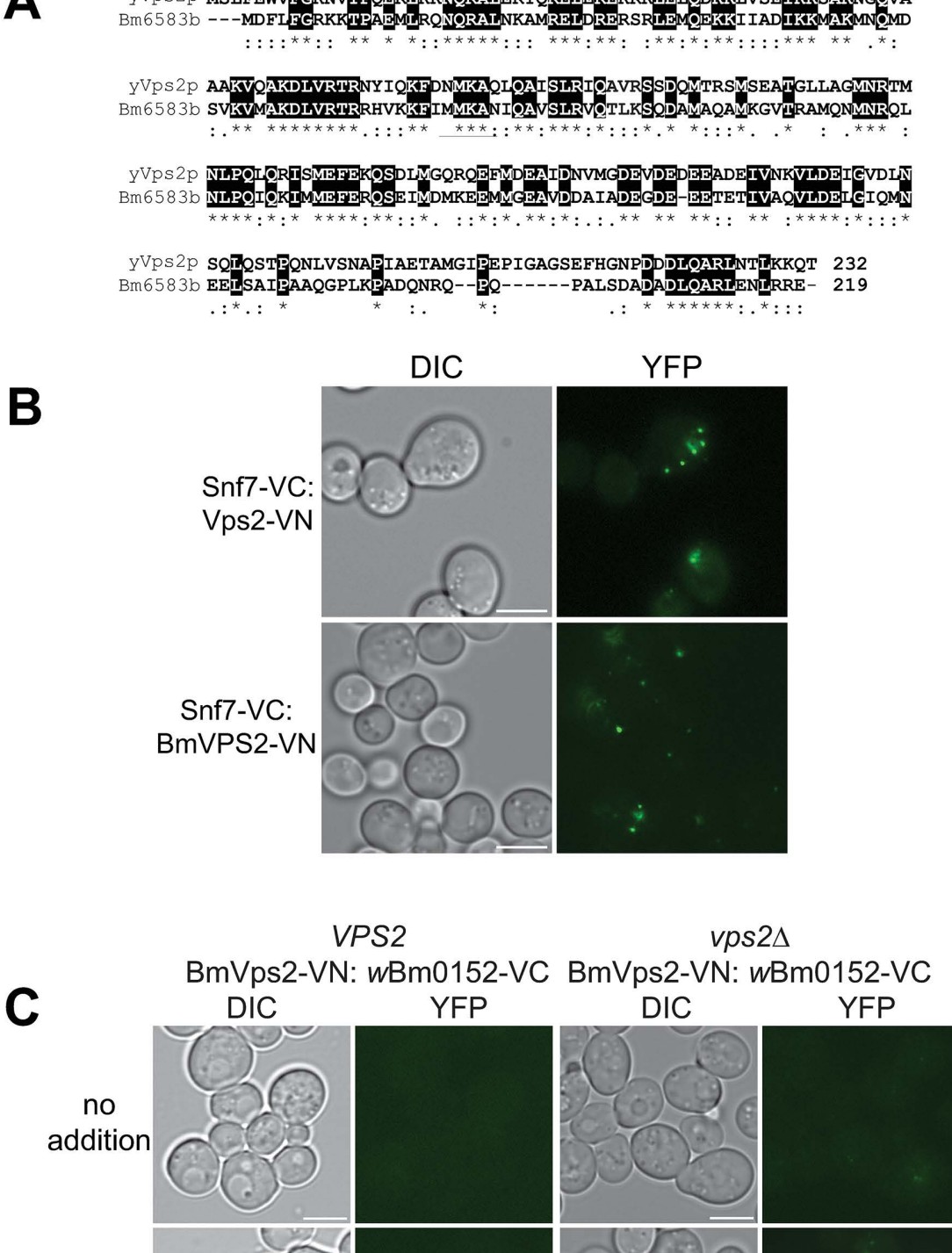

**Fig 8. Wbm0152 binds *Brugia malayi* Vps2 ortholog (Bm6583b) in yeast.** A) Sequence alignment of Vps2 protein from yeast (yVps2) and *Brugia* Vps2 homolog (*Bm*Vps2) in which highlighted residues (*) indicate identical amino acids, (:) indicating highly similar amino acids and (.) indicating similar residues. B) Yeast strain SEY6210 expressing Snf7-VC and harboring either the pRS414-pVPS2-*VPS2*-VN (top) or pRS414-pVPS2-*BmVPS2*-VN expression vector were grown in CSM media lacking uracil and tryptophan for 18h at 30° C with shaking, diluted 1:10 into fresh selective media, and outgrown for 6 hours before imaging. C) Yeast strain SEY6210 harboring yeast wild type SEY6210 (left) or *vps2*Δ (right) strains harboring the plasmids pRS414-pVPS2-*Bm6583b*-VN (Methods) and pYES*CUP1*-*w*Bm0152-VC were grown as in B), except fresh selective media either contained or lacked 0.5 mM CuSO4 during outgrowth. Bar = 5 μ; images are representative of three separate experiments.

embryos. Perhaps *w*Bm0152 overexpression and subsequent manipulation of ESCRT is important for initial entry of *Wolbachia* into new cells within the nematode via disruption of membrane budding pathways.

Interestingly, Wbm0152 belongs to a conserved family of peptidoglycan-associated lipoproteins (Pal) found extensively throughout Gram-negative bacteria, where it plays a role in bacterial outer membrane stability and in cell division [22,101]. In Gram-negative bacteria, Pal proteins interact with the peptidoglycan cell wall, outer membrane proteins, and the periplasmic TolB protein – which is part of the larger inner membrane Tol complex (consisting of TolQ, TolR, and TolA) – and concentrates at the site of cell division to modulate peptidoglycan processing during cell division via the recruitment of cell separation amidases [24,102]. While it is difficult to hypothesize how a periplasmic-facing outer membrane lipoprotein could interact with 'external' host cell proteins in the context of *Wolbachia* physiology, it is important to note that the *Wolbachia* genome does not appear to contain any known Tol protein homologs – including TolB [103]– suggesting the role of this Pal-like protein in *Wolbachia* may be quite different than typically observed in most Gram-negative bacteria. Furthermore, Pal-family proteins have been shown in some bacteria to have a 'dual confirmation' in which the population of Pal can have an 'inward' facing (periplasmic) C-terminus, or an 'outward' facing, exposed C-terminus on the outer membrane surface. Despite this well documented phenotype, the physiological role of this 'outward' facing Pal population remains unknown [104]. In support of the hypothesis that Wbm0152 is outward facing in *Wolbachia*, Wbm0152 has been previously localized to both the surface of the *Wolbachia* bacterium – as well as on the membrane of the *Wolbachia*-containing vacuole – via immunofluorescence and immuno-EM in *Brugia* tissues [105]. The fact that Wbm0152 is present on the surface of the *Wolbachia*-containing vacuole suggests that this protein may be positioned to interact directly with *Brugia* cytoplasmic proteins, like ESCRT.

Why might *Wolbachia* modulate ESCRT activity in *Brugia*? As *Wolbachia* lives inside of a membrane-bound compartment within nematode cells, we could hypothesize *Wolbachia* may be modulating *Brugia* ESCRT function to initiate ILV formation into the *Wolbachia*-containing compartment, perhaps to deliver nutrients from the cytosol of the host cell, or to increase the accumulation of *Brugia*-cytosolic lipid droplets resulting from the inhibition of ESCRT. Furthermore, this activity could also be utilized either for the creation of the *Wolbachia*-containing vacuole from post-Golgi membranes, or to prevent entry into lysosomal degradation pathways that require ESCRT activity, such as autophagy. Lastly, it is known that *Wolbachia* is required to both maintain a quiescent pool of germline stem cells in the female *Brugia* germline and stimulates mitotic cell division during embryonic development [106]. As ESCRT-III activity was also found to be critical for abscission during cytokinesis and nuclear envelope repair during cell division [31,107,108], it is even possible that *Wolbachia* directly controls the cell cycle of the *Brugia* germline cells through the manipulation of ESCRT-III activity. Unfortunately, direct experiments to test these hypotheses are difficult without greatly expanding the availability of research reagents for both *Brugia* and *Wolbachia*. Nevertheless, this study marks an important milestone in the analysis of the molecular relationship between *Wolbachia* and the filarial nematode host, opening doors for new drug development for the eradication of *Wolbachia* – and subsequently – filarial nematodes.

## Methods

### Yeast strain and plasmid constructions

All microscopy-based experiments were performed with derivatives of the yeast strain SEY6210 (*MATa ura3–52 leu2–3, 112 his3-Δ100 trp1-Δ901 lys2–801 suc2-Δ9*). For the growth experiments performed in Figs 2 and 7, yeast strains were

derivatives of BY4742 (MATα his3Δ1 leu2Δ0 lys2Δ0 ura3Δ0). Any expression studies utilizing β-estradiol for the induction of *GAL1* promoters required transforming relevant yeast strains with linearized pAGL (a gift from Dr. Daniel Gottschling, University of Washington), thus introducing the gene encoding for the Gal4-estrogen receptor-VP16 (GEV) chimeric protein into the *leu2* locus [109].

To create a high-copy, constitutive yeast expression vector for *w*Bm0152, we replaced the *GAL1* promoter contained in pYES2/NT A (ThermoFisher Scientific) with the strong, constitutive *TDH3* promoter. The *TDH3* promoter was amplified from the p413-GPD plasmid [110] using the primer pair YTDH F and YTDH R (S1 Table), each containing 30-bp of homology upstream and downstream of the *GAL1* promoter in pYES2/NT A. The resultant amplicon was co-transformed into BY4742 with pYES2/NT A that had previously been linearized with PvuII, using standard lithium acetate techniques [111]. Gap-repaired plasmids were selected on CSM medium lacking uracil. To generate the pYES$_{TDH3}$-*w*Bm0152 plasmid, the pYES2-*w*Bm0152 plasmid [17] was digested with PmeI and HindIII, the *w*Bm0152-containing insert was gel purified and co-transformed with pYES-*TDH3* previously linearized with BamHI, as above.

To construct the galactose-inducible *Escherichia coli pal* expression vector, the *pal* open reading frame was amplified from the *E. coli* K12 chromosome with the primer pair EcPAL F and EcPAL R, containing 30-bp sequences homologous to pYES2, in-frame with the N-terminal Xpress epitope. The resultant amplicon was co-transformed into *S. cerevisiae* BY4742 with the pYES2/NT A vector, previously linearized with BamHI, creating pYES-*Ec*PAL. To generate the Snf7-mRuby plasmid used in Fig 4, the *SNF7* ORF was amplified from the *S. cerevisiae* BY4742 genome using primer pair YSNF7 F and YSNF7 R. This amplicon contained 300-bp sequences both upstream and downstream of *SNF7*, containing the *SNF7* promoter and terminator regions. This amplicon was co-transformed into *S. cerevisiae* BY4742 with the pYES2/NT A vector, previously linearized with BamHI, creating pYES-P$_{SNF7}$-*SNF7*. Then, the mRuby ORF was amplified from the previously-created pYES-*w*Bm0076-mRuby vector [21] with the primer pair SNF7-mRubyF and SNF7-mRuby R, amplifying the mRuby ORF with homologous sequences to *SNF7* and the pYES backbone. This amplicon was co-transformed with the pYES-P$_{SNF7}$-*SNF7* vector, previously linearized with PmeI, creating pYES-P$_{SNF7}$-*SNF7*-mRuby.

To create a high-copy, inducible yeast expression vector for *w*Bm0152 that does not rely on activation of the *GAL1* promoter, we replaced the *GAL1* promoter from pYES2/NT A with the copper-inducible *CUP1* promoter [112]. To avoid repetitive DNA sequences contained in the duplicated *S. cerevisiae CUP1–1 CUP1–2* locus, we first amplified the upstream sequence of *CUP1–1* with the primer pair CUP F and CUP R. The resultant amplicon was used as template with the primer pair YCUP F and YCUP R; the amplicon from this second reaction was then co-transformed into yeast with pYES2/NT A previously linearized with PvuII. To create the *w*Bm0152 expression construct in this background, pYES$_{CUP1}$ was linearized with BamHI and co-transformed with the PmeI-HindIII *w*Bm0152-containing insert from above.

The copper-inducible pYES$_{CUP1}$-*w*Bm0152 split Venus yeast expression plasmids used for the biomolecular fluorescence complementation assays were created by amplifying either the VN or VC domain from the relevant plasmid template using the indicated primer pairs containing 30-bp homology to the C-terminus of *w*Bm0152 and the pYES plasmid backbone (VN: pFA6a-VN-HIS3 with 0152VCN F and 0152VN R; VC: pFA6a-VC-TRP1 with 0152VCN F and 0152VC R) [69]. The resultant amplicons were separately co-transformed into yeast with PmeI-linearized pYES$_{CUP1}$-*w*Bm0152 to create the final constructs via gap repair. To generate the pYES$_{CUP1}$-*VPS2*-VN split Venus yeast expression plasmid, the VPS2-VN fusion was amplified off the genome of the SEY6210 *VPS2*-VN yeast strain [45], using the primer pair VPS2VN F and VPS2VN R. The resultant amplicon was co-transformed into yeast with BamHI-linearized pYES$_{CUP1}$-*w*Bm0152 to create the final construct via gap repair.

The *Brugia malayi VPS2* homolog was identified via blastp [113] using the yeast Did4p/Vps2p sequence and restricting the search to *Brugia malayi*. The closest match was *Bm6583b* (GenBank accession: VIO93588) and the transcript cDNA encoding for this ORF was retrieved from the WormBase web site, http://www.wormbase.org, release WS296, date Apr 30 2025 [114]. This sequence was codon-optimized for *Saccharomyces cerevisiae* expression using the Codon Optimization Tool (IDT Technologies). When generating the *Bm6583b* gBlock (IDT Technologies), we also appended the 300 base

pairs immediately upstream of the yeast *VPS2* ORF, containing the endogenous promoter region. Prior to the *Bm6583b* stop codon, we added sequences encoding for the c-Myc epitope for detection via immunoblot, a BamHI restriction site immediately after the stop codon, and the 300 bases immediately downstream of the yeast *VPS2* ORF, containing the endogenous terminator region. Finally, we appended 30-bp sequences to the 5' and 3' ends of this sequence, providing homology flanking the MCS of the yeast plasmid pRS414 for cloning via gap repair in yeast (S1 Table). This gBlock was co-transformed into yeast with pRS414 previously linearized with BamHI, creating pRS414-*Bm6583*-myc. To generate the *Bm6583* expression plasmid used for the split Venus bimolecular fluorescence complementation study, the Venus VN domain was amplified from pFA6a-VN-HIS3 with the primer pair *BM6583* VN F and *BM6583* VN R (S1 Table) and co-transformed into yeast with pRS414-*Bm6583*-myc plasmid previously digested with BamHI.

All generated plasmid constructs were confirmed through whole plasmid sequencing by Plasmidsaurus using Oxford Nanopore Technology with custom analysis and annotation.

### RapIDeg protein turnover assay

To visualize the ESCRT-dependent turnover of Fth1p in response to rapamycin, we modified the RapIDeg yeast strain Fth1-GFP-FKBP 3x Ub (SEY6210.1 *tor1−1 fpr1Δ::NATMX6* pRS305-pGPD-FRB-3xUb::*LEU2* FTH1-GFP-2xFKBP::*HIS3*, a gift from Dr. Ming Li, University of Michigan) [51] to express *GAL* promoters with β-estradiol. First, we amplified the *HPHMX6* gene from pAG32 [115] using primer pair pr29 and pr32 (S1 Table); this amplicon contains homology to the *NATMX6* cassette inserted into the *FPR1* gene. The RapIDeg strain was transformed with this amplicon and hygromycin-resistant colonies were selected. This strain was then screened for nourseothricin sensitivity, confirming the replacement of the *NATMX6* cassette with *HPHMX6*. The resultant strain was then transformed with the linear pAGL, as above, creating the β-estradiol-responsive RapIDeg strain.

RapIDeg strains harboring either the galactose-inducible pYES-*w*Bm0152 plasmid or empty vector control were grown to saturation in selective medium at 30° C for 16h. Strains were subcultured into 10 mL fresh selective medium and grown to mid-log for 4h at 30 °C. To induce the expression of *w*Bm0152, 1 µM β-estradiol was added to each culture and incubated at 30° C with shaking for another 2h. After 2h, each culture was evenly split and either rapamycin or the DMSO vehicle control was added to 1 µg mL$^{-1}$ and cultures were outgrown for an additional 2h at 30° C. 1 mL aliquots were removed from each condition, yeast cells were harvested by centrifugation (7000 x *g*, 1 min), and visualized via fluorescence microscopy. From the remaining culture, 3.0 OD$_{600}$ units were harvested via centrifugation and total cellular proteins were extracted from cell pellets [116] for immunoblotting experiments.

### Microscopy

Cells to be visualized were grown overnight at 30° C in selective media, subcultured in fresh media with or without induction agent(s), and grown for an additional 5 hours. Cells were harvested via centrifugation, washed with sterile water, and suspended in 50 µL water. Cell suspensions were mounted to slides pre-treated with a 1:1 mixture of polylysine (10% w/v):concanavalin A (2 mg mL$^{-1}$) solution. Cells were visualized using a Nikon Ti-U fluorescence microscope, and images were processed using the Fiji software package (ImageJ2, v2.14.0/1.54f) [117,118]. Colocalization analyses were performed using the Coloc2 plugin contained within the Fiji package.

### Thin section electron microscopy

*Saccharomyces cerevisiae* strain SEY6210 *GAL*$^+$ harboring either a vector control or the pYES-*w*Bm0152 expression plasmid were grown in selective media (CSM lacking uracil) for 18h at 30° C with shaking. Cells were harvested via centrifugation, washed with sterile water, then diluted 1:10 into fresh CSM-ura containing 2% galactose to induce *w*Bm0152 expression. Cells were harvested at log phase, collected on 0.45µ filter discs (Millipore) via vacuum, loaded into 0.25 mm aluminum hats, and high-pressure frozen in the Wolwend high-pressure freezing machine (HPF) as previously described

[58]. Freeze substitution was carried out with media containing 0.1% uranyl acetate and 0.25% gluteraldehyde in anhydrous acetone [119] in a Leica AFS (Automated Freeze Substitution, Vienna, Austria). Samples were then embedded in Lowicryl HM20, and UV-polymerized at -60C over 2 weeks (Polysciences, Warrington, PA, A. Staehelin, personal communication, Wemmer et al JCB 2011). A Leica Ultra-Microtome was used to cut 80 nm serial thin sections and 200 nm serial semi-thick sections from polymerized blocks and sections were collected onto 1% formvar films adhered to rhodium-plated copper grids (Leica Biosystems, Nussloch, Germany, and Electron Microscopy Sciences). Thin sections were imaged using a FEI Tecnai T12 "Spirit" electron microscope (120 kV, AMT 2 x 2 k CCD).

### Semi-thick section electron tomography

Semi-thick sections of polymerized blocks (~200 nm) were applied to grids labeled on both sides with fiduciary 15 nm colloidal gold (British Biocell International). Dual-axis tilt series were collected of the samples from +/- 60 degrees with 1-degree increments at 300kV using SerialEM [120] on a Tecnai 30 FEG electron microscope (FEI-Company, Eindhoven, the Netherlands). Tilt series were regularly imaged at 23,000X using SerialEM [121], with 2X binning when recording on a 4x4K CCD camera (Gatan, Inc., Abingdon, UK) creating a 2x2K image with a pixel size of 1.02 nm. Tomograms were constructed and modeled with the IMOD software package (3dmod 4.0.11, [122]). All membrane structures were identified with areas of interest modeled (ER-LD compartments, small MVBs, tubular MVBs). IMODINFO provided surface area and volume data of contour models, diameters and distances were measured from the outer membrane leaflets at optimal X Y orientations in the tomograms at 50 nm intervals using 3DMOD.

### Statistical analysis

Statistical analysis was performed within the Prism software package (GraphPad Software, v. 10.4.0). Column statistics were performed via a 1-way ANOVA Dunnett's multiple comparison test with single pooled variance. Where noted in figures, ns = $P > 0.05$ (not significant); (*) = $P \le 0.05$; (**) = $P \le 0.01$; (****) = $P \le 0.0001$.

### Supporting information

**S1 Fig. *Ec*PAL expressing yeast strains are not sensitive to Congo red. A)** Sequence alignment of *w*Bm0152 compared to *E. coli* homolog *Ec*PAL in which highlighted residues (*) indicate identical amino acids, (:) indicate highly similar amino acids and (.) indicate similar residues between the two proteins. **B)** BY4742 strains containing either a beta estradiol inducible pYES-*Ec*PAL, pYES-*w*Bm0152 or empty vector control strain grown overnight shaking at 30° C, then subcultured into 5mL of CSM-URA media with and without 1 μM beta estradiol for four hours to induce expression. Whole cell lysates were collected and probed with anti-Xpress antibody to detect indicated protein and anti-Sec18 as a loading control. **C)** Strains were grown overnight at 30° C in CSM-URA media. Cultures were diluted to a final $OD_{600} = 1.0$, serially diluted 10-fold four times into sterile water, and 10 μL of each dilution was spotted to CSM media lacking uracil either lacking or containing 15 μg mL$^{-1}$ Congo red. Plates were incubated at 30° C and imaged after 72h.
(TIF)

**S2 Fig. CryoEM technique preserves lipid membranes.** Cells from the wild type strain, SEY6210, were imaged at 200 nM (left), 50 nm (middle), and 4 nm (right), thicknesses of the tomographic volume, to visualize various membrane bound organelles. Organelles denoted as nucleus (N), mitochondria (M), endoplasmic reticulum (ER), and autophagosome (*).
(TIF)

**S1 Movie. Representation of tomographic space used to model intracellular endomembrane compartments in SEY6210 harboring a vector control strain grown in the presence of 2% galactose.** These images were used to generate Fig 4A–4C; bar = 100 nm.
(MP4)

**S2 Movie. Representation of tomographic space used to model intracellular endomembrane compartments in SEY6210 harboring the galactose-inducible *w*Bm0152 expression vector grown in the presence of 2% galactose.** These images were used to generate Fig 4D–4F; bar = 100 nm.
(MP4)

**S3 Fig. *w*Bm0152 does not interact with other ESCRT-III subunits in vivo.** SEY6210 strains harboring either the N-terminus (VN) or C-terminus (VC) of a Venus-YFP molecule on the C-terminus of the indicated ESCRT subunit were transformed with the corresponding copper-inducible pYES$_{CUP1}$-*w*Bm0152-VN or pYES$_{CUP1}$-*w*Bm0152-VC plasmid. Strains were grown in CSM media lacking uracil for 18h at 30° C with shaking, diluted 1:10 into fresh selective media lacking or supplemented with 0.5 mM CuSO$_4$, and outgrown for 6 hours before imaging. Bar = 5 µ; images are representative of three separate experiments.
(TIF)

**S1 Table. Primer and nucleotide sequences used in this study.**
(DOCX)

## Acknowledgments

The authors would like to thank Dr. Ming Li (University of Michigan) for providing the yeast RapIDeg strain used in these studies and Dr. Christian Ungermann (University of Osnabrück) for providing the *MNN9*-GFP yeast strain. The authors would also like to thank Anne Nguyen for excellent technical assistance.

## Author contributions

**Conceptualization:** Lindsay Berardi, Vincent J. Starai.

**Data curation:** Lindsay Berardi, Matthew West.

**Funding acquisition:** Greg Odorizzi, Vincent J. Starai.

**Investigation:** Lindsay Berardi, Alora Colvin, Matthew West.

**Methodology:** Lindsay Berardi, Matthew West, Vincent J. Starai.

**Project administration:** Vincent J. Starai.

**Supervision:** Greg Odorizzi.

**Writing – original draft:** Lindsay Berardi, Vincent J. Starai.

**Writing – review & editing:** Lindsay Berardi, Matthew West, Greg Odorizzi, Vincent J. Starai.

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
