## [Decision Letter · Decision Letter 0]

31 Aug 2025

Wbm0152, an outer membrane lipoprotein of the *Wolbachia* endosymbiont of *Brugia malayi* , inhibits yeast ESCRT complex activity

PLOS Pathogens

Dear Dr. Starai,

Thank you for submitting your manuscript to PLOS Pathogens. After careful consideration, we feel that it has merit but does not fully meet PLOS Pathogens's publication criteria as it currently stands. Therefore, we invite you to submit a revised version of the manuscript that addresses the points raised during the review process.

Please submit your revised manuscript within 60 days Oct 30 2025 11:59PM. If you will need more time than this to complete your revisions, please reply to this message or contact the journal office at plospathogens@plos.org. Please include the following items when submitting your revised manuscript:

We look forward to receiving your revised manuscript.

Kind regards,

Mostafa Zamanian

Academic Editor

PLOS Pathogens

Tracey Lamb

Section Editor

PLOS Pathogens

Editor-in-Chief

PLOS Pathogens

orcid.org/0000-0003-2946-9497

Michael Malim

PLOS Pathogens

orcid.org/0000-0002-7699-2064

**Journal Requirements:**

- ® on pages: 23, and 45

- TM on page: 2.

4) We notice that your supplementary Figure, and Table are included in the manuscript file. Please remove them and upload them with the file type 'Supporting Information'. Please ensure that each Supporting Information file has a legend listed in the manuscript after the references list.

5) In the online submission form, you indicated that "All data used in this submission can be freely obtained by contacting the corresponding author." All PLOS journals now require all data underlying the findings described in their manuscript to be freely available to other researchers, either

1. In a public repository

2. Within the manuscript itself

3. Uploaded as supplementary information.

Note : Authors must share the “minimal data set” for their submission. PLOS defines the minimal data set to consist of the data required to replicate all study findings reported in the article, as well as related metadata and methods (https://journals.plos.org/plosone/s/data-availability#loc-minimal-data-set-definition).

**Reviewers' Comments:**

Reviewer's Responses to Questions

**Part I - Summary**

Reviewer #1: In this manuscript, the authors take advantage of the yeast ectopic expression in order to determine the potential in vivo functions of the outer membrane lipoprotein wBm0152 contained in the genome of the Wolbachia present in the filarial nematode B. malayi. The authors demonstrate that ectopic expression of wBm0152 inhibits yeast ESCRT activity and defect endosomal maturation. In addition fluorescent assays are used to demonstrate that wBm0192 associates with Vps2p, a subunit of the ESCRT-III complex. Given the technical restrictions working with Wolbachia and the dearth of functional insight into Wolbachia surface proteins and effectors, this is valuable contribution to the field not only for the insights into wBm0152 function bur also because it provides a general framework for future functional studies. Once published this will be of interest to those studying Wolbachia, filarial nematode and more generally symbiosis. Having said this, the following issues should be addressed prior to publication

Reviewer #2: This manuscript describes the phenotypes, in yeast, upon expression of a Wolbachia outer membrane lipoprotein. The authors do a good job of introducing the system and explaining the ESCRT pathway and overall, explaining assays throughout and their limitations. I think it would be beneficial if we had a table or diagram of all the ESCRT subunits used in this study so that it is clearer to readers which subunit is in which complex (especially because a lot of readers do not have a strong cell biology background). Overall, I found the study to be interesting, although I have several concerns, which I list below.

Reviewer #3: The manuscript titled "Wbm0152, an outer membrane lipoprotein of the Wolbachia endosymbiont of Brugia malayi, inhibits yeast ESCRT complex activity" investigates the interaction between a Wolbachia protein with the yeast Endosomal Sorting Complex Required for Transport (ESCRT). The study demonstrates that expressing a Wolbachia outer membrane lipoprotein, wBm0152, in Saccharomyces cerevisiae significantly disrupts endosomal maturation and affects ubiquitylated protein turnover. Through in vivo bimolecular fluorescence complementation, the authors discovered that Wbm0152 interacts with the Vps2p subunit of the ESCRT-III subcomplex blocking further bacterial degradation through the host endosomal pathway. They also tested for interactions with the ortholog (BmVps2, Bm6583b) from the Wolbachia host nematode, Brugia malayi. These authors suggest a role for ESCRT in Wolbachia persistence, potentially shedding light on the complex relationship between Wolbachia and its filarial host. Since ESCRTs are crucial for various cellular processes like endosomal maturation, autophagy, and cell division, this interesting study proposes a potential mechanism by which Wolbachia might manipulate Brugia's membrane trafficking to ensure its survival within host cells.

Considering that Wolbachia of filaria cannot be cultured and that molecular studies in the filaria can be a bit challenging, the advantage of using the yeast model is that it allows much easier molecular manipulation “in vivo” and so mechanisms can be tested. While largely well designed, this study provides indirect evidence of wBm0152’s potential effect on its host. For the authors to state that these data suggest a novel role of ESCRT in Wolbachia persistence would require further validation in the filarial host. The yeast system, albeit a powerful model, is very different from a nematode system.

**Part II – Major Issues: Key Experiments Required for Acceptance**

Reviewer #1: 1. Can the authors modify and quantify the extent of the over-expression? I do have a concern that some of the effects may be a general response of expressing large amounts of a foreign protein. This is especially a concern in the localization studies. Is the localization the same at different levels of expression? As a control, the authors may want to consider expressing scrambled version of wBM 0152

2. ESCRT has a number of diverse roles in yeast including nuclear envelope reformation and cytokinesis abscission. What is the effect of over-expression on these events?

3. It would be helpful to include mitochondria, nuclear membrane images to determine the quality of membrane preservation in the EM images

4. Because wBm0152 is a membrane bound protein, rather that an effector protein free in the host cytoplasm, there is some concern regarding interpretation of the results. The ectopically expressed Wolbachia membrane-associated domains of wBm0152 are available to associate with host organelles unlike the in vivo situation. While expressing only the non-membrane associated is beyond the scope of the manuscript, this caveat should be considered in the discussion.

Reviewer #2: 1) Because this work is necessarily focused on heterologous assays, controls are especially important. The authors are appropriately circumspect about what they can infer/conclude from these assays. That said, on lines 398-402 they attempt to address the fact that Pal proteins are periplasmic-facing lipoproteins so it is unclear how they would interact with host cell proteins. I think an excellent control for the importance of this result would be to express E.coli Pal proteins in yeast and confirm that they do not have the same phenotype. This would strengthen the argument that Wolbachia Pal proteins are behaving differently.

2) Does Congo Red only interact with ESCRT mutants? Are there other cellular processes that might be impacted leading to these results? Along the same lines, is the accumulation of lipid droplets and the deformation of the ER normal for ESCRT mutants? Or is this just a side effect of the effector? Could this be relevant to other things that wBm0152 is doing in the cell?

3) Throughout – it would be good to have quantification, especially for the microscopy figures. Additionally, statistical analyses for all figure panels would be good to include. Also, Figure 4 needs better controls – what is weak or strong colocalization? Perhaps quantification would help to assess this. The Pearson correlation coefficient is sensitive to background and differences in intensities between two channels. By having a positive and negative control for these experiments, it makes the colocalization of the effector and subunits more meaningful to the reader. For Figure5, why are there so many fluorescent puncta in the uninduced control? Quantification of the puncta will help strengthen their argument.

4) Line 242-244: “These results show a reduction in the recruitment of Bro1p to assembled ESCRT complexes in the presence of Wbm0152” Can you make that assertion based on colocalization of Bro1 and wBm0152? You would need to measure colocalization of Bro1 with the ESCRT machinery in the presence and absence of the effector.

5) Line 268-272: “Interestingly, we did not see 269 reconstitution of Venus fluorescence in Vps2-VC/wBm0152-VN strains (Fig. S1). This observation could be due to the C-terminal -VN fusion forcing Vps2 into a constitutively active, ‘open’ conformation to bind Wbm0152, which is known to occur with yeast and mammalian ESCRT-III GFP fusion proteins.” Are we worried that some of the other subunits are in the wrong confirmation then? Is it possible that some of the negative results shown in Fig S1 are a result of being on the wrong terminus? Only one orientation of Vps25, Vps20, Snf7, Bro1, and Doa4 are shown.

6) Figure 6 - It would be helpful to add the complexes that each subunit is in onto the figure to make the order of assembly clearer to readers. Vps2 is the one subunit that you show to physically interact with wBm0152, but you cannot show that it is necessary for wBm0152 activity. Is there some orthogonal approach you can use to show that this interaction is important to the activity?

7) Figure 7. It is nice to show that the effector interacts with a host protein, but it would also be nice to have some sort of in vivo connection since all of this is out of the natural context. They make guesses about how this might be beneficial to the Wolbachia, how does the effector’s activity actually affect the bacterium? I appreciate that this system is very difficult to work in, but are there ESCRT complex inhibitors that can be given to nematodes to assess effects on Wolbachia titer? Or perhaps some coIP in the natural system? I think this would help solidify the significance of the finding.

Reviewer #3: 1. While yeast serves as a convenient model, it seems that one important experiment missing is validation of these findings in the natural host, Brugia malayi. The absence of direct validation in a nematode limits the conclusions. As the authors mentioned in the Discussion, RNAi on host genes of ESCRT in JW (Drosophila cells) resulted in an increase of insect Wolbachia. A similar experiment (RNAi on proteins of the ESCRT complex) on Brugia would help validate the results obtained in the yeast model. The validation would also support the functional essentiality of the ESCRT-III in Brugia and its physiological role in controlling symbiosis. These are pretty substantial experiments so at this point the best would likely be to tamper the message.

For example, line 136-139 states: “These findings not only identify a bacterial protein capable of inhibiting ESCRT activity but also illuminate an intricate relationship between Wolbachia and its nematode host to support its intracellular survival.” The correct statement should be that what this demonstrates is that a bacterial protein can inhibit ESCRT activity in yeast. As for illuminating the intricate relationship between Wolbachia/filaria, that seems an overstatement.

2. I’m a bit surprised that the authors used an empty vector as a control. Wouldn’t a better control be an unrelated protein with a similar length to wBm0152? My worry is that in yeast, even an unrelated protein could potentially have the same effect. Please clarify why you don’t think that would be the case.

The manuscript also does not provide sufficient details on the statistical analysis of the data. For instance, the criteria used to quantify colocalization or the statistical significance of differences in growth defects and protein degradation assays are not clearly described. Details on the number of replicates and the statistical tests used should be provided to ensure robustness.

3. Discussion: The study concludes that wBm0152 interferes with ESCRT to support Wolbachia persistence in the host. However, the manuscript does not explore the physiological consequences of this interaction in the context of Wolbachia's lifecycle within Brugia malayi. For example, it is known that Wolbachia numbers increase exponentially during worm development when the worm enters a mammalian host. It will be beneficial to discuss how bacteria manage this expansion. Can Wolbachia overexpress wBm0152 during the development of the worms? Transcriptomic data of Wolbachia in worms at different stages are available, and the data would help support the discussion. It would be beneficial to discuss how the ESCRT-Wbm0152 interaction influences Wolbachia's intracellular survival, replication, or transmission in the nematode host.

**Part III – Minor Issues: Editorial and Data Presentation Modifications**

Reviewer #1: 1. Lines 158-162- I'm confused because the percentages listed in the text do not match the percentages listed in Figure 1

2. Typo in abstract: "These mosquito borne pathogens......". The vector for river-blindness is the Black fly not a mosquito

3. Lines 337-339 "Through a combination of mutant insect....the importance of host actin dynamics for the uptake and maternal transmission of Wolbachia......." I suggest the literature taken as a whole demonstrate that host microtubules play a dominant role in Wolbachia's subcellular interactions. This should be considered as there is a growing literature on physical and functional interactions between ESCRT and microtubules

Reviewer #2: The bulk of the discussion was a nice summary of what is known about intracellular bacteria interacting with the ESCRT complex, what is known about this effector and others like it, and how this work fits into the field as a whole.

Line 325-326: “we have shown that expression of Wbm0152 in yeast inhibits the eukaryotic ESCRT complex via interactions with the core ESCRT-III protein, Vps2p.” Actually, they show that the effector interacts with Vps2, and they show that the effector inhibits the ESCRT complex, but they don’t actually show that the vps2 interaction is what causes the inhibition. They couldn’t determine that with their Congo Red assay, so they can’t really assert this.

As stated above, a discussion of the lipid accumulations and perturbed ER morphology upon wBm0152 expression would be appreciated. How can this be related to the ESCRT complex? Or is there another cellular process that is affected? Are there any known ER stress responses known to occur in Brugia with Wolbachia (maybe in that RNA seq study that is referenced)?

Reviewer #3: • Authors used yeast deletion strains in this study; could the overexpression of ESCRT-III rescue wBm0152 effects? This could be part of follow-up studies that confirm the competitive aspect of Wbm0152 interactions with ESCRT-III components.

• Fig 1: It would help the reader to add labels to the fluorescent pictures, pointing to the vacuolar membrane and to the vacuole lumen, as described in the results.

• Additionally, the listing of the percentages on the fluorescent images is confusing when also referring to the percentage of cells (2%) accumulating Fth1-GFP in the vacuole lumen 2h after treatment with rapamycin (lines 162-163). One percentage refers to the number of cells that express fluorescence (vacuole membrane or lumen), while the other one refers to the remaining wBm0152 cells that still degrade Fth1-GFP after Rapamycin treatment. Since the percentages of ‘no rapamycin’ minus ‘rapamycin’ comes also to 2% in both the vector control and the wBm0152 cells, it becomes confusing.

• The authors mention many proteins in their complexes, and it is at time confusing. A schema of these proteins in ESCRT-III and the potential localization of wBm0152 would be helpful to the reader.

• The authors refer to Wolbachia pipientis – this nomenclature comes from the fact that the first Wolbachia discovered was in the mosquito Culex pipientis. However, the nomenclature of Wolbachia is a bit messy and so, in the Wolbachia field, Wolbachia are generally referred to by the name of their host. For example, Wolbachia of Drosophila melanogaster is wMel; Wolbachia of Brugia malayi (which is very different genomically from Wolbachia of arthropods) is referred to as wBm. I suggest the authors take out “pipientis” when referring to Wolbachia throughout their manuscript.

• The introduction section would benefit from more detailed information about the significance of ESCRT complexes in eukaryotic cells and their known interactions with intracellular pathogens.

• The discussion section could be expanded to include a more thorough comparison of the study's findings with existing literature on ESCRT manipulation by other intracellular pathogens.

• What are the advantages of understanding ESCRT in bacterial pathogenesis? Any future directions?

• Abstract: Line 19: “These mosquito-borne pathogens…” Onchocerca parasites are transmitted by black flies, not by mosquitoes.

• At the end of the abstract, I suggest the authors add a statement, like they did in the author summary, on why looking at ESCRT is important.

• Line 98: Should be “led” not “lead”

• Line 147-148 “the Rapamycin Dependent Degradation Assay (RapIDeg)” before the authors stated it as “line 128: the rapamycin-induced degradation (RapIDeg) system”.

• Lines 204-205. As ILVs and MVBs are measured in the following Figure, some explanation on the physiological connections between ILV and MVB would help better understand the results.

• Line 267: If the snf7p is not colocalized with wBm0152, how do the authors explain their colocalization in the previous result (Fig. 4)?

• Results starting at Line 302: In that first paragraph, you mention the Brugia ortholog. Showing % homology between Brugia vps2 and yeast vps2 would be helpful.

• Line 361: What is the physiological role(s) of ESCRTs in cell biology?

• G should be capitalized in Gram-negative throughout the text.

• The TEM pictures should be labeled.

PLOS authors have the option to publish the peer review history of their article (what does this mean? ). If published, this will include your full peer review and any attached files.

**Do you want your identity to be public for this peer review?** For information about this choice, including consent withdrawal, please see our Privacy Policy .

Reviewer #1: No

Reviewer #2: No

Reviewer #3: No

**Figure resubmission:**

**Reproducibility:**



---

## [Editor Report · Decision Letter 1]

4 Dec 2025

Dear Dr. Starai,

I believe your revision addresses the primary points in the first round of review and I am pleased to inform you that your manuscript 'Wbm0152, an outer membrane lipoprotein of the *Wolbachia* endosymbiont of *Brugia malayi* , inhibits yeast ESCRT complex activity' has been provisionally accepted for publication in PLOS Pathogens. 

Best regards,

Mostafa Zamanian

Academic Editor

PLOS Pathogens

Tracey Lamb

Section Editor

PLOS Pathogens

Sumita Bhaduri-McIntosh

Editor-in-Chief

PLOS Pathogens

orcid.org/0000-0003-2946-9497

Michael Malim

Editor-in-Chief

PLOS Pathogens

orcid.org/0000-0002-7699-2064
---

## [Editor Report · Acceptance letter]

Dear Dr. Starai,

We are delighted to inform you that your manuscript, "Wbm0152, an outer membrane lipoprotein of the *Wolbachia* endosymbiont of *Brugia malayi* , inhibits yeast ESCRT complex activity," has been formally accepted for publication in PLOS Pathogens.

Best regards,

Sumita Bhaduri-McIntosh

Editor-in-Chief

PLOS Pathogens

orcid.org/0000-0003-2946-9497

Michael Malim

Editor-in-Chief

PLOS Pathogens

orcid.org/0000-0002-7699-2064